# The Impact of Postoperative Renal Function Recovery after Laparoscopic and Robot-Assisted Partial Nephrectomy in Patients with Renal Cell Carcinoma

**DOI:** 10.3390/medicina58040485

**Published:** 2022-03-27

**Authors:** Kota Kawase, Torai Enomoto, Makoto Kawase, Manabu Takai, Daiki Kato, Shota Fujimoto, Koji Iinuma, Keita Nakane, Seiichi Kato, Noriyasu Hagiwara, Masahiro Uno, Takuya Koie

**Affiliations:** 1Department of Urology, Gifu University Graduate School of Medicine, Gifu 5011194, Japan; stnf55@gifu-u.ac.jp (K.K.); buki2121@gifu-u.ac.jp (M.K.); takai_mb@gifu-u.ac.jp (M.T.); andreas7@gifu-u.ac.jp (D.K.); kiinuma@gifu-u.ac.jp (K.I.); keitaco@gifu-u.ac.jp (K.N.); 2Department of Urology, Matsunami General Hospital, Gifu 5016062, Japan; G.T.E.24150107@gmail.com (T.E.); hagiwara44@gmail.com (N.H.); 3Department of Urology, Ogaki Municipal Hospital, Ogaki 5038502, Japan; uro2@omh.ogaki.gifu.jp (S.F.); seikatou@xj.commufa.jp (S.K.); mmys@octn.jp (M.U.)

**Keywords:** renal cell carcinoma, robot-assisted partial nephrectomy, laparoscopic partial nephrectomy, postoperative renal function, trifecta

## Abstract

*Background and objectives:* This study aimed to evaluate the association between warm ischemic time (WIT) and postoperative renal function using Trifecta achievement in patients with renal cell carcinoma (RCC) who underwent robotic (RAPN) or laparoscopic partial nephrectomy (LPN). *Materials and Methods:* We conducted a retrospective multicenter cohort study of patients with RCC who underwent RAPN (RAPN group) or LPN (LPN group) at three institutions in Japan between March 2012 and October 2021. The primary endpoints were the rate of trifecta achievement in both surgical techniques and the association between WIT and recovery of postoperative renal function surgical outcomes. *Results:* The rate of trifecta achievement was significantly lower in patients with LPN than in those with RAPN (*p* < 0.001). WIT ≥ 25 min were 18 patients (18%) in the RAPN group and 89 (52.7%) in the LPN group. The postoperative estimated glomerular filtration rate (eGFR) was almost the same. However, 13 patients (7.7%) had a decreased in eGFR ≥ 15% at 3 months after LPN compared with the preoperative eGFR. *Conclusions:* The rate of trifecta achievement in the RAPN group was significantly higher than that in the LPN group. However, eGFR was identified as relatively better preserved after PN in both groups.

## 1. Introduction

To date, several guidelines recommend nephron-sparing surgery for small renal masses, such as T1 renal cell carcinoma (RCC), to preserve the normal renal parenchyma and avoid the increased risk of chronic renal disease, as well as obtaining oncological efficacy via radical nephrectomy [1,2]. For many years, open partial nephrectomy (PN) has been the standard surgical treatment for small RCC. Recently, minimally invasive surgery, including robot-assisted PN (RAPN) or laparoscopic PN (LPN), has been adopted worldwide, thereby contributing to comparable oncological outcomes, less morbidity, and shortened convalescence compared to open PN [3,4,5]. The main objective of PN is to completely remove the renal mass to preserve renal function without causing surgical complications [6]. “Trifecta” and “Pentafecta” are widely adopted due to simplification and standardization of the comparison of the outcomes of PN [7,8]. Namely, the accomplishment of trifecta was defined as a negative surgical margin, less surgery-related complications, and warm ischemia time (WIT) of ≤25 min [7]. Pentafecta is defined as the absence of an upstage of chronic kidney disease and a minimum of 90% total estimated glomerular filtration rate (eGFR) preservation, in addition to trifecta [8]. According to both composite outcome measures, WIT is an important factor, having no small effect on the recovery of postoperative renal function [3].

Based on data from basic research, a duration of 30 min was considered as an optimal threshold for a long time [9]. LPN is a technically challenging procedure; therefore, WIT was significantly longer than RAPN based on seven nonrandomized observational studies that enrolled >300 RAPN and >400 LPN patients [10]. In contrast, hilar tumors were associated with longer WIT and operating time than non-hilar tumors in patients who underwent RAPN [11]. Recently, several authors have reevaluated whether WIT itself is the cause of renal function loss or whether WIT is dependent on tumor or surgical factors and might have limited impact on renal function [3,12].

Therefore, this study aimed to evaluate the association between WIT and postoperative renal function using trifecta achievement in RCC patients who underwent RAPN or LPN.

## 2. Materials and Methods

### 2.1. Patients

This study was approved by the Institutional Review Board of Gifu University (authorization number: 2021-052; date: 12/May/2021) and the institutional review boards. Patient consent was not required owing to the retrospective nature of the study. The provisions of the ethics committee and ethics guidelines in Japan did not require written consent because the study information was disclosed to the public in the case of retrospective and/or observational studies like this one, using materials such as existing documentation. The details of the study can be accessed at http://www.med.gifu-u.ac.jp/file/2020-271.pdf (accessed on 13 January 2022).

We conducted a retrospective multicenter cohort study of patients with RCC who underwent RAPN (RAPN group) or LPN (LPN group) at Gifu University Hospital, Matsunami General Hospital, and Ogaki Municipal Hospital in Japan between March 2012 and October 2021. All LPNs in this study were performed by expert surgeons who had experience of at least 20 cases with small renal tumors. Preoperative information included patient age, height, weight, tumor side, tumor size, clinical stage, R.E.N.A.L nephrometry score [13], serum creatinine (Cr), and the preoperative eGFR. Peri- and postoperative outcomes were collected as follows: surgical approach, operative time, WIT, estimated blood loss (EBL), number of renal arteries, blood transfusion rate, pathological T stage, histopathological type of the resected tumor, surgical margin status, and perioperative complications. Postoperative serum Cr level and eGFR were assessed at one day, two weeks, and one, three, and six months.

eGFR was calculated using the Modification of Diet in Renal Disease 2 equation, further modified for Japanese patients as outlined by the Japanese Society of Nephrology (eGFR = 1.94 × serum Cr mg/dL 1.094 × age × [0.739 if female] [14]. All tumors were staged according to the American Joint Committee on Cancer’s eighth edition cancer staging manual [15]. Surgery-related perioperative complications were evaluated according to the Clavien–Dindo classification [16]. Trifecta was calculated using three variables: surgical margin status, WIT (<25 min), and perioperative complications based on the Clavien-Dindo classification [17].

### 2.2. Subsection

Both RAPN and LPN via trans- or retroperitoneal approaches were performed using previously described techniques [18]. LPN procedures were performed in four ports, whereas RAPN was performed using four ports for the da Vinci device and two assistant ports in the transperitoneal approach or one port in the retroperitoneal approach. A total renal arterial clamping technique was used in all cases. To determine the excision margins, we used an ultrasound probe to examine the location, depth, and borders of renal tumors. Renal masses were resected at a renal parenchyma margin of at least 1 mm with RAPN and 3mm with LPN. After extirpation of the renal tumor, an inner running suture using absorbable sutures was applied to close the visible bleeding vessels, sinus, or collecting system. If necessary, a running suture using a barbed suture with a sliding clip technique was performed on the renal parenchyma with absorbable hemostatic [19].

### 2.3. Endpoints and Statistical Analysis

The primary endpoints were the rate of trifecta achievement in both surgical techniques and the association between WIT and recovery of postoperative renal function surgical outcomes. JMP 14 (SAS Institute Inc., Cary, NC, USA) was used for the data analyses. Continuous variables were compared using the Student’s *t*-test, and categorical variables were compared using the Fisher’s exact or the McNemar tests. Two-sided *p*-values were calculated, and the significance level was set at *p* < 0.05.

## 3. Results

### 3.1. Patient Characteristics

Patient demographic data are presented in Table 1. The median age, BMI, and follow-up period of the enrolled patients were 65 years, 24.0 kg/m^2^, and 35.5 months, respectively. The BMI in the RAPN group was significantly higher than that in the LPN group.

### 3.2. Surgical and Pathological Outcomes

Surgical outcomes of the enrolled patients are shown in Table 2. Both RAPN and LPN were performed using the transperitoneal approach in 158 patients (58.7%). Although the operative time was significantly shorter in the RAPN group than in the LPN group, EBL was significantly lower in the RAPN group than in the LPN group. However, only one patient received blood transfusion.

Pathological findings of the enrolled patients are listed in Table 3. Eleven (6.5%) patients were diagnosed with pathological T3a disease.

At the end of the follow-up period, one patient with T1b and a negative surgical margin died of RCC because of multiple bone and lymph node metastases. Five patients (3.0%) in the LPN group were alive with recurrence or distant metastases (lung metastasis in two patients and liver metastasis, bone metastasis, and local recurrence in one patient each), and no patient in the RAPN group had local recurrence or distant metastasis.

### 3.3. Trifecta and the Chronological Change of Postoperative Renal Function

The trifecta analysis results are shown in Table 4. According to the rate of trifecta achievement, LPN patients had a significantly lower rate than did those with RAPN. Twenty-five patients (14.8%) had positive surgical margins, including clear cell carcinoma in 17, papillary RCC in three, and chromophobe RCC in one. According to the postoperative complications, seven (7%) and two (2%) patients in the RAPN group and 14 (8.3%) and 3 (1.8%) in the LPN group had Clavien-Dindo surgical complications grade ≤II and III, respectively (Table 5). WIT ≥ 25 min were observed in 18 (18%) and 89 (52.7%) patients in the RAPN and LPN groups, respectively. WIT was the most strongly associated with trifecta achievement.

The chronological changes in renal function are shown in Figure 1. Although the WIT was significantly longer in the LPN group than in the RAPN group, the postoperative eGFR was almost the same. However, 13 patients (7.7%) had a decreased eGFR ≥ 15% at three months after LPN compared with the preoperative eGFR, although no patients had decreased eGFR before and after RAPN.

## 4. Discussion

To date, RAPN and LPN have been recommended as minimally invasive surgeries compared to open PN [20]. Of these, RAPN allows enhanced precision with a shortened surgical learning curve and shorter operative and ischemic times, with less EBL compared with LPN [20]. Tachibana et al. reported that RAPN had significantly better preservation of renal function than LPN [18]. Recently, RAPN has expanded to include highly complex and clinical T2 renal tumors [21]. Nonetheless, the rate of trifecta was relatively lower than that in reports from RAPN for small renal masses with increasing tumor complexity and size [22,23]. Trifecta provides an indication of surgical quality in PN patients [8]. In previous studies, the rate of trifecta achievement ranged between 31.6% and 87.8% [4,6,7,8]. For these reasons, surgical experience and approach may have been associated with the outcomes of trifecta [8]. Regarding the surgical approach, RAPN was more likely to achieve trifecta than LPN, even though Mehra et al. reported no significant differences in the rate of trifecta achievement after open PN, LPN, and RAPN [4,8]. Owing to three-dimensional vision and wrist articulation, RAPN has enabled the detection of relatively challenging renal tumors with less difficulty, and more often [8]. RAPN had 6.8-times higher odds of achieving Trifecta than LPN [8]. However, the rate of trifecta achievement increased significantly with the surgeon and hospital volume according to the analysis of 1222 PNs from 11 institutions [24]. In this study, the rate of positive surgical margins in the LPN group was relatively high. This rise in the positive margin rate may have been caused by factors including the surgeons’ experience, or the surgical procedure for tumor resection. Therefore, surgeon volume was associated with positive surgical margins, a longer WIT, and the operative time, whereas the major complication rate was associated with hospital volume [24].

Although whether renal ischemia independently impacts the renal function preservation or is merely dependent on tumor complexity or volume remains debated [25], WIT might have identified a key determinant of nephron damage from PN, along with the removal of healthy nephrons and damage from the renorrhaphy technique [3,12,18]. Tachibana et al. reported that preservation of a greater volume of renal parenchyma and a shorter ischemic time may avoid decreasing renal function [18]. Zhang et al. reported an association between prolonged ischemia time and the development of acute kidney injury [26]. Conversely, WIT ≥30 min was not associated with an increased risk of chronic kidney disease or any major renal function deterioration, up to 5 years postoperatively, from a propensity score-matched analysis of 1816 patients who had undergone PN [27]. Several studies have identified that parenchymal volume loss and higher baseline eGFR are significant predictors of postoperative renal function, whereas WIT has a relatively smaller impact [28,29]. Factors including surgeon experience, precise and/or meticulous dissection of the normal renal parenchyma, and minimizing the damage from renorrhaphy may be more important than the duration of WIT in preserving the renal function after PN [30]. In this study, 10 patients in the RAPN group and 26 patients have a R.E.N.A.L. nephrometry score of nine or higher. There were no significant differences in the rates of decline and recovery of renal function after surgery between the two surgical approaches, although LPN had a significantly longer WIT than that of RAPN. However, 7.7% of the patients who had WITs ≥30 min had eGFRs ≥15% at 3 months after LPN.

Our study had some limitations. First, it was a retrospective study involving multicenter data; thus, it was susceptible to potential bias owing to diagnostic and therapeutic differences among participating institutions. Second, it had a relatively small sample size and short follow-up period. Therefore, a longer observation period is necessary for oncological consequences. Third, we did not investigate pentafecta due to missing data 12 months after surgery. Finally, the surgical procedure was performed at the discretion and/or experience of the primary physician.

## 5. Conclusions

The rate of trifecta achievement was significantly higher in the RAPN than in the LPN group. Although WIT was not associated with postoperative renal function in this study, WIT ≥ 30 may have a potential predictor for renal function after PN. In addition, the parenchymal volume of the kidney may be necessary to maintain postoperative renal function. Therefore, RAPN may not only provide safe removal of the renal tumor, good cancer control, and prevention of postoperative complications, but also the preservation of renal function. In addition, RAPN may have potential advantages regarding the treatment of large, multiple, hilar renal masses, or renal tumor with high R.E.N.A.L. score.

## Figures and Tables

**Figure 1 medicina-58-00485-f001:**
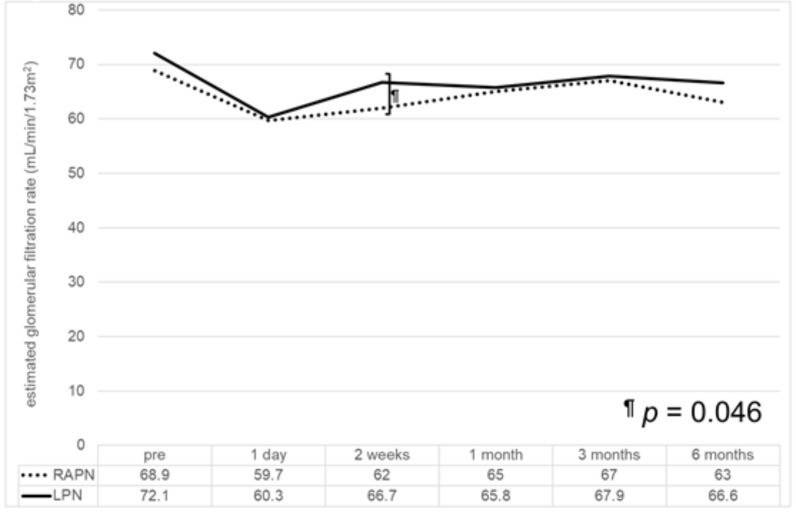
The chronological changes in the renal function after robot-assisted (RAPN) and laparoscopic partial nephrectomy (LPN). At two weeks after surgery, the estimated glomerular filtration rate in the LPN group was significantly better than that in the RAPN group. However, there were no statistically significant differences in both groups at one day, and one, three, and six months after surgery.

**Table 1 medicina-58-00485-t001:** Patient characteristics.

	RAPN	LPN	*p*
Patients (number)	100	169	
Age (year, median, Interquartile range)	65 (55–73)	64 (56–72)	0.815
Gender (number, %)			0.563
Male	76 (76)	123 (72.8)	
Female	24 (24)	46 (27.2)	
Body mass index (kg/m^2^, median, Interquartile range)	24.9 (21.9–27.2)	23.5 (21.2–26.1)	0.042
Tumor side (number, %)			0.717
Right	52 (52)	84 (49.7)	
Left	48 (48)	85 (50.3)	
Tumor size (mm, median, Interquartile range)	25 (19–35)	26 (18–35)	0.703
Clinical T stage (number, %)			0.896
T1a	85 (85)	143 (84.6)	
T1b	14 (14)	26 (15.4)	
T2a	1 (1)	0	
R.E.N.A.L. nephrometry score (number, median, Interquartile range)	6 (5–8)	6 (5–8)	0.850
Follow-up period (months, median, Interquartile range)	18.9 (12–34.7)	46.9 (26.1–70.9)	<0.001

RARP; robot-assisted partial nephrectomy: LPN; laparoscopic partial nephrectomy.

**Table 2 medicina-58-00485-t002:** Surgical outcomes.

	RAPN	LPN	*p*
Patients (number)	100	169	
Surgical approach (number, %)			<0.001
Transperitoneal	75 (75)	83 (49.1)	
Retroperitoneal	25 (25)	86 (50.9)	
Operative time (min, median, interquartile range)	200 (167–240)	233 (183–279)	<0.001
EBL (mL, median, interquartile range)	50 (10–150)	20 (10–100)	0.041
Renal artery (number, interquartile range)	1 (1–2)	1 (1–1)	0.464
Blood transfusion (number, %)	1 (1)	0	0.194
Postoperative complications (number, %)	13 (13)	19 (11.2)	0.699
Clavien-Dindo classification (number, %)			0.813
Grade II	15 (15)	21 (12.4)	
Grade III	2 (2)	3 (1.8)	

RAPN; robot-assisted partial nephrectomy: LPN; laparoscopic partial nephrectomy: EBL; estimated blood loss.

**Table 3 medicina-58-00485-t003:** Pathological outcomes.

	RAPN	LPN	*p*
Patients (number)	100	169	
Pathological T (number, %)			0.762
T1a	87 (87)	150 (88.8)	
T1b	8 (8)	12 (7.1)	
T2a	1 (1)	0	
T3a	4 (4)	7 (4.1)	
Histopathological type (number, %)			0.566
Clear cell	83 (83)	139 (82.2)	
Papillary	10 (10)	24 (14.2)	
Chromophobe	7 (7)	6 (3.6)	

RAPN; robot-assisted partial nephrectomy: LPN; laparoscopic partial nephrectomy.

**Table 4 medicina-58-00485-t004:** Trifecta analysis comparing RAPN and LPN.

Outcome	RAPN	LPN	* p *
Trifecta achievement (number, %)	74 (74)	67 (39.6)	<0.001
WIT (min, median, IQR)	18 (16–23)	25 (19–31)	<0.001
Negative surgical margin (number, %)	96 (96)	152 (89.9)	0.074
No complications (number, %)	91 (91)	152 (89.9)	0.194

RAPN = robot assisted partial nephrectomy. LPN = laparoscopic partial nephrectomy. WIT = warm ischemic time. IQR = interquartile range.

**Table 5 medicina-58-00485-t005:** Perioperative complications.

Type of Complication (Number, %)	RAPN (*n* = 100)		LPN (*n* = 169)	
	Grade II	Grade III	Grade II	Grade III
Infection	2 (2)	0	4 (2.7)	0
Ileus	3 (3)	0	0	0
Postoperative hemorrhage	0	0	3 (1.8)	0
Postoperative urine leakage	0	0	1 (0.6)	1 (0.6)
Depression	1 (1)	0	0	0
Cerebral infarction	1 (1)	0	0	0
Hypertension	0	0	1 (0.6)	0
Diabetes Mellitus	0	0	1 (0.6)	0
Liver function deterioration	0	0	2 (1.2)	0
Adrenal insufficiency	0	0	1 (0.6)	0
Pneumothorax	0	0	1 (0.6)	0
Ventricular fibrillation	0	1 (1)	0	0
Rapture of pseudoaneurysm	0	1 (1)	0	0
Postoperative renal death	0	0	0	1 (0.6)
Necrotic cholecystitis	0	0	0	1 (0.6)

RAPN = robot assisted partial nephrectomy. LPN = laparoscopic partial nephrectomy. WIT = warm ischemic time.

## Data Availability

The data presented in this study are available on request from the corresponding author. The data are not publicly available due to privacy and ethical reasons.

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
