# Peer review of "The Impact of Postoperative Renal Function Recovery after Laparoscopic and Robot-Assisted Partial Nephrectomy in Patients with Renal Cell Carcinoma"

_medicina, 2022, doi:10.3390/medicina58040485_

Round 1

Reviewer 1 Report

In this manuscript, the authors conducted a retrospective multicenter cohort study of patients with RCC who underwent RAPN (RAPN group) or LPN (LPN group) at three institutions in Japan. The primary endpoints included the rate of trifecta achievement in both surgical techniques while considering the association between WIT and surgical outcomes, particularly the recovery of postoperative renal function

The study showed that patients with LPN had significantly lower rates of trifecta achievement than those with RAPN (p < .001). Moreover, WIT ≥ 25 min was observed in 18 (18%) and 89 (52.7%) patients in the RAPN and LPN groups, respectively. However, the postoperative eGFR was virtually the same. The authors therefore concluded that the rate of trifecta achievement in the RAPN group was significantly higher than that in the LPN group. However, their findings revealed that eGFR was relatively better preserved after PN in both groups. Despite some limitations of this paper, I believe it provides substantial information, especially for Japanese urologists.

Minor comments; 

I could not find Figure 4. Please comment about this.

Author Response

22, March, 2022

Dear Editor-in-Chief, the Medicina

Thank you very much for the opportunity to resubmit our manuscript. The comments of the reviewers have been helpful in allowing us to revise our manuscript according to the reviewers’ recommendations. It would be our great pleasure if you would take our manuscript into consideration for publication in the Medicina.

I look forward to your reply.

Best regards,

Takuya Koie, M.D.

Professor

Department of Urology, Gifu University Graduate School of Medicine

Japan

Response to Reviewer 1

The authors appreciate the reviewer’s comments. The authors’ point-by-point responses to the comments are given below.

  1. In this manuscript, the authors conducted a retrospective multicenter cohort study of patients with RCC who underwent RAPN (RAPN group) or LPN (LPN group) at three institutions in Japan. The primary endpoints included the rate of trifecta achievement in both surgical techniques while considering the association between WIT and surgical outcomes, particularly the recovery of postoperative renal function.

The study showed that patients with LPN had significantly lower rates of trifecta achievement than those with RAPN (p < .001). Moreover, WIT ≥ 25 min was observed in 18 (18%) and 89 (52.7%) patients in the RAPN and LPN groups, respectively. However, the postoperative eGFR was virtually the same. The authors therefore concluded that the rate of trifecta achievement in the RAPN group was significantly higher than that in the LPN group. However, their findings revealed that eGFR was relatively better preserved after PN in both groups. Despite some limitations of this paper, I believe it provides substantial information, especially for Japanese urologists.

Minor comments;

I could not find Figure 4. Please comment about this.

Response:

The authors have added the Table 4 on line 150.

Reviewer 2 Report

This study was investigated the clinical data from the patients treated with robot-assisted partial nephrectomy and laparoscopic partial nephrectomy in Japanese multi-centers. The outcomes focused on peri-operative data which may be used daily clinical practice.

Several issues, if addressed, would further improve the manuscript.

Minor

Line 52 RARP→RAPN

Line161 RARP→RAPN

Major

1.Table 4 is missing. The details are unknown.

2.The authors note that WIT of 25 minutes or more was observed in 18% of patients in the RAPN group and in 52.7% of patients in the LPN group, but the changes in renal function after surgery were almost the same, the authors note. In fact, renal function after surgery may be affected more by the loss of normal renal parenchyma due to resection than by the time of ischemia.

Although more complex tumors usually require more than 25 minutes of ischemia, they account for only a small portion of the total number of cases in this study, 10 in the RAPN group and 26 in the LPN group.

It is considered that the technique and experience of the operator greatly affect the reason why WIT is extended.

What is the number of cases experienced by surgeons in this study?

It is considered that the factor of the operator greatly influences in LPN in which the learning curve becomes long. It is necessary to describe this point.

3.The document states that the margin is at least 1 mm when the tumor is removed.

Is the excision procedure the same for both LPN and RAPN?

If only the sharp dissection is used, a 1 mm resection margin with LPN is technically difficult.

A margin of 1 mm is usually considered to be an enucleoresection procedure.

The positive margin rate in the LPN group in this study was 14.8%, which is considered high. Is this caused by the excision procedure?

Because the surgical procedure involves much of the loss of normal renal parenchyma, consideration is needed.

Author Response

22, March, 2022

Dear Editor-in-Chief, the Medicina

Thank you very much for the opportunity to resubmit our manuscript. The comments of the reviewers have been helpful in allowing us to revise our manuscript according to the reviewers’ recommendations. It would be our great pleasure if you would take our manuscript into consideration for publication in the Medicina.

I look forward to your reply.

Best regards,

Takuya Koie, M.D.

Professor

Department of Urology, Gifu University Graduate School of Medicine

Japan

Response to Reviewer 2

The authors appreciate the reviewer’s comments. The authors’ point-by-point responses to the comments are given below.

This study was investigated the clinical data from the patients treated with robot-assisted partial nephrectomy and laparoscopic partial nephrectomy in Japanese multi-centers. The outcomes focused on perioperative data which may be used daily clinical practice.

Several issues, if addressed, would further improve the manuscript.

Minor

  1. Line 52 RARP→RAPN

Line161 RARP→RAPN

Response:

The authors have revised this term.

Major

1.Table 4 is missing. The details are unknown.

Response:

The authors have added the Table 4 on line 150.

2.The authors note that WIT of 25 minutes or more was observed in 18% of

patients in the RAPN group and in 52.7% of patients in the LPN group, but

the changes in renal function after surgery were almost the same, the

authors note. In fact, renal function after surgery may be affected more by

the loss of normal renal parenchyma due to resection than by the time of

ischemia.

Although more complex tumors usually require more than 25 minutes of ischemia, they account for only a small portion of the total number of cases in this study, 10 in the RAPN group and 26 in the LPN group.

It is considered that the technique and experience of the operator greatly affect the reason why WIT is extended.

What is the number of cases experienced by surgeons in this study?

It is considered that the factor of the operator greatly influences in LPN in which the learning curve becomes long. It is necessary to describe this point.

Response:

The authors have added the following sentence on line 73:

All LPNs in this study were performed by expert surgeons who had an experience of at least 20 cases with small renal tumors.

3.The document states that the margin is at least 1 mm when the tumor is

removed.

Is the excision procedure the same for both LPN and RAPN?

If only the sharp dissection is used, a 1 mm resection margin with LPN is

technically difficult.

A margin of 1 mm is usually considered to be an enucleoresection

procedure.

The positive margin rate in the LPN group in this study was 14.8%, which is considered high. Is this caused by the excision procedure?

Because the surgical procedure involves much of the loss of normal renal parenchyma, consideration is needed.

Response:

The authors have revised the following part on line 97:

Renal masses were resected at a renal parenchyma margin of at least 1 mm with

RAPN and 3 mm with LPN.

The authors have revised the following sentence on line 189:

In this study, the rate of positive surgical margins in the LPN group was relatively high. This rise in the positive margin rate may have been caused due to factors including the surgeons’ experience, or the surgical procedure for tumor resection.

The authors have already described the following sentence on the lines 191 and

202:

Tachibana et al. reported that preservation of a greater volume of renal

parenchyma and a shorter ischemic time may avoid decreasing renal function

[18].

In this study, 10 patients in the RAPN group and 26 patients have a R.E.N.A.L.

nephrometry score of nine or higher. There were no significant differences in the

rates of decline and recovery of renal function after surgery between the two

surgical approaches, although LPN had a significantly longer WIT than that of

RAPN. However, 7.7% of the patients who had WITs ≥30 min had eGFRs ≥15%

at 3 months after LPN.

Reviewer 3 Report

This study by Kawase et al. focused on evaluating the impact of Trifecta achievement on renal function in patients with renal cell carcinoma (RCC) who underwent robotic surgery (RAPN) or laparoscopic partial nephrectomy (LPN). The data is solid and is presented in an organized manner. However, following points should be addressed:

  1. Isn't the word “RARP” used in line 52 a mistake for RAPN?
  2. The authors mention Table 4, but Table 4 is not listed.
  3. A comparison of the frequency of complications affecting renal function (eg. hypertension and diabetes) in each group would add more weight to the results.

Author Response

22, March, 2022

Dear Editor-in-Chief, the Medicina

Thank you very much for the opportunity to resubmit our manuscript. The comments of the reviewers have been helpful in allowing us to revise our manuscript according to the reviewers’ recommendations. It would be our great pleasure if you would take our manuscript into consideration for publication in the Medicina.

I look forward to your reply.

Best regards,

Takuya Koie, M.D.

Professor

Department of Urology, Gifu University Graduate School of Medicine

Japan

Response to Reviewer 3

The authors appreciate the reviewer’s comments. The authors’ point-by-point responses to the comments are given below.

This study by Kawase et al. focused on evaluating the impact of Trifecta

achievement on renal function in patients with renal cell carcinoma (RCC)

who underwent robotic surgery (RAPN) or laparoscopic partial

nephrectomy (LPN). The data is solid and is presented in an organized

manner. However, following points should be addressed:

  1. Isn't the word “RARP” used in line 52 a mistake for RAPN?

Response:

The authors have revised this point.

  1. The authors mention Table 4, but Table 4 is not listed.

Response:

The authors have added the Table 4 on line 150.

  1. A comparison of the frequency of complications affecting renal function

(eg. hypertension and diabetes) in each group would add more weight to

the results.

Response:

The authors did not collect the data with preoperative comorbidities, including

hypertension or diabetes mellitus. The authors have added the Table 5 on line

155.

Round 2

Reviewer 2 Report

This second version of the paper is a great improvement, the authors are to be commended
The manuscript has been revised well. I think this manuscript will be acceptable.